A survey of researchers’ methods sharing practices and priorities

LaFlamme Marcel 1 mlaflamme@plos.org
Harney James 1
Hrynaszkiewicz Iain 2
1 Public Library of Science , San Francisco, CA , United States
2 Public Library of Science , Cambridge , United Kingdom
Gray Andrew
Electronic publication date: 2024 Jan 3
Publication date: 2024
Volume: 12
Electronic Location ID: e16731
Received 2023 Jul 11; Accepted 2023 Dec 6
Copyright: © 2024 LaFlamme et al.
Copyright year: 2024
Copyright holder: LaFlamme et al.
License: This is an open access article distributed under the terms of the Creative Commons Attribution License, which permits unrestricted use, distribution, reproduction and adaptation in any medium and for any purpose provided that it is properly attributed. For attribution, the original author(s), title, publication source (PeerJ) and either DOI or URL of the article must be cited.
License URL: https://creativecommons.org/licenses/by/4.0/

Keywords: Methods sharing, Protocol sharing, Research methods, Open science

Funding: The authors received no funding for this work.

==============================
Missing or inaccessible information about the methods used in scientific research slows the pace of discovery and hampers reproducibility. Yet little is known about how, why, and under what conditions researchers share detailed methods information, or about how such practices vary across social categories like career stage, field, and region. In this exploratory study, we surveyed 997 active researchers about their attitudes and behaviors with respect to methods sharing. The most common approach reported by respondents was private sharing upon request, but a substantial minority (33%) had publicly shared detailed methods information independently of their research findings. The most widely used channels for public sharing were connected to peer-reviewed publications, while the most significant barriers to public sharing were found to be lack of time and lack of awareness about how or where to share. Insofar as respondents were moderately satisfied with their ability to accomplish various goals associated with methods sharing, we conclude that efforts to increase public sharing may wish to focus on enhancing and building awareness of existing solutions—even as future research should seek to understand the needs of methods users and the extent to which they align with prevailing practices of sharing.

Introduction

The production of credible scientific knowledge involves the reporting of not only results, but also the methods by which they were attained. By the second half of the twentieth century, English-language scientific journals across a range of fields had adopted a standard structure for research articles including a separate Methods section (Sollaci & Pereira, 2004), in which information about materials, instruments, and procedures was to be presented. Yet concerns about the adequacy of this information, especially given space constraints linked to print distribution, soon gave rise to publication formats that were independent of the findings of a particular study, such as the recipe-style protocols compiled in specialized book series (Walker, 1984). The transition to digital distribution brought new methods journals and repositories in the early years of the twenty-first century (Teytelman & Ganley, 2021), although most of these products and services remained highly field-specific.

Against the backdrop of mounting concerns around reproducibility (Fanelli, 2018) and the science reform movement that these catalyzed (Field, 2022), a more general attention to issues of methods reporting emerged. Guidelines for reporting, which originated with clinical trials research in the 1990s, were extended to other study types (Altman & Simera, 2016) and adopted, if unevenly, by authors and editors (Fuller et al., 2015). Persistent identifiers were introduced to help address ambiguities around the use of reagents, tools, and materials (Bandrowski et al., 2016). Meanwhile, calls to routinely and publicly share detailed descriptions of experimental procedures anticipated gains in research transparency and efficiency (Crotty & Stebbins, 2021) and were even codified in the UNESCO Recommendation on Open Science (UNESCO, 2021a). Given the absence of widely accepted standards for such descriptions, though, formats have proliferated and efforts to enable structured search across them (Giraldo et al., 2017) have gained little traction. Funders have also been slower to implement specific policies on methods sharing than other Open Science practices like data sharing (but see (ASAP, 2021; Secretariat for National Open Science and Research Coordination, 2023)), such that uptake by researchers has thus far been limited.

At the scientific publisher PLOS, formal efforts to promote methods sharing began in 2017 with guidance encouraging authors to deposit their laboratory protocols at protocols.io, a platform for developing and sharing reproducible methods (PLOS, 2017). This partnership deepened in 2020 with the launch of Lab Protocols, an innovative article type consisting of a step-by-step protocol posted to protocols.io and a complementary, peer-reviewed publication in PLOS ONE (Hrynaszkiewicz, 2020). As of this writing, more than 100 Lab Protocols have been published, mostly but not exclusively in the life sciences. The proportion of research articles that link out to protocols on protocols.io peaked in 2018 but has since fallen off, amounting to less than 1% of PLOS’s published output (Public Library of Science, 2023).

Existing research has shown that missing (Glasziou et al., 2008; Haddaway & Verhoeven, 2015) or inaccessible (Standvoss et al., 2022) methods information costs researchers time and hampers reproducibility. Yet little is known about how, why, and under what conditions researchers share detailed methods information, or about how such practices vary across social categories like career stage, field, and region. Factors blocking methods sharing are said to include time burden and loss of competitive advantage (Crotty & Stebbins, 2021), but empirical evidence has yet to be gathered from the sharer’s perspective. The present study aims to explore researchers’ attitudes and behaviors with respect to methods sharing, while also helping PLOS and other stakeholders to optimize existing methods sharing solutions and identify opportunities for further innovation.

Materials and Methods

To inform the design of a survey questionnaire, semi-structured preliminary interviews were conducted with 12 researchers from a range of research fields. Interviewees had publicly shared detailed methods information in a peer-reviewed publication or on a methods sharing platform in the past 5 years. Written notes from these interviews were consulted in creating the questionnaire, which was programmed in the online survey tool Alchemer.

The structure of the survey questionnaire was adapted from previous research at PLOS on researchers’ Open Science attitudes and behaviors (Hrynaszkiewicz, Harney & Cadwallader, 2021b). Respondents were asked to provide demographic information and then to answer a series of questions about their practices of and attitudes toward sharing detailed methods information independently of their research findings, i.e., outside of the Methods section of a research article.

To identify potential opportunities for supporting researchers in sharing methods, respondents were asked to rate a series of goals related to methods sharing on a five-point ordinal scale. Each goal was rated separately for its importance and the satisfaction of the respondent with their ability to achieve the goal. These responses were then converted into numerical importance and satisfaction scores, drawing on the Jobs to be Done framework for understanding consumer action (Ulwick, 2016). A score of 0 indicates that the respondent regards the goal as not at all important or is completely dissatisfied with their ability to achieve the goal. A score of 100 indicates that the respondent regards the goal as extremely important or is completely satisfied with their ability to achieve the goal. The intermediate points on the ordinal scale are assigned values of 25, 50, and 75 respectively.

Respondents were also asked to rate a series of potential features of a product or service intended to enable the public sharing of detailed methods information in terms of their value to the respondent. Most of these features were derived from PLOS’s Lab Protocol article type, although they were not identified as such in the questionnaire.

A draft of the questionnaire was sent to four interviewees to elicit feedback on the survey’s length, structure, and clarity of item design. After minor wording changes were implemented, the survey (LaFlamme, Harney & Hrynaszkiewicz, 2023) was deployed on 30 March 2022. Recruitment of respondents included email invitations to PLOS authors and protocols.io users, as well as researchers connected to TCC Africa, a training center that provides support to scientists across Africa and is a partner of PLOS. Invitations were also posted to relevant listservs and on Twitter. The results of the survey were exported from Alchemer on 19 May 2022.

Data analysis

Statistical analysis was primarily descriptive in nature. To explore correlations between practices of and attitudes toward sharing detailed methods information, the Wilcoxon-Mann-Whitney test was used to compare responses by respondents who had publicly shared detailed methods information and respondents who had not. A multiple test correction was also applied using the Benjamini-Hochberg procedure. The uneven distribution of respondents by career stage, field, and region limited our ability to compare responses across these segments except in an exploratory way; the numbers of respondents in some categories were too small to be compared with a high degree of confidence.

Ethical considerations

The Heartland Institutional Review Board granted approval for this study (HIRB Project No. 07092023-494) and classified it as entailing no more than minimal risk to participants. Heartland IRB also waived the need for written informed consent. Participants were informed that their participation in the survey was completely voluntary, and that they were free to withdraw from the study at any point until submitting their response. The data collection procedures and survey questionnaire were compliant with the General Data Protection Regulation 2016/679.

Results

The survey yielded 1,514 total responses; given the study’s recruitment strategy, it was not possible to calculate a response rate. Responses from individuals who did not consider themselves to be active scientific researchers (n = 142) and partial responses from individuals who did not progress to the end of the survey (n = 375) were excluded. Differences between partial and completed responses were explored; the distribution of responses by field was similar, while a greater proportion of partial responses were from researchers at earlier career stages (46% vs. 37% for completed) and researchers in regions other than North America or Europe (46% vs. 38% for completed). Responses from the 997 individuals who completed the survey—those who progressed to the end and answered most or all questions—are reported in Table 1.

Table 1 Completed responses by career stage, field, and region.

Career stage	Number of responses	% of all responses	
Has not (yet) been awarded terminal degree	87	9%	
0–5 years since terminal degree	154	15%	
6–10 years since terminal degree	128	13%	
11–15 years since terminal degree	150	15%	
16–20 years since terminal degree	105	11%	
20+ years since terminal degree	372	37%	
Field			
Biology and life sciences	435	44%	
Ecology and environmental sciences	75	8%	
Engineering and technology	21	2%	
Humanities and social sciences	62	8%	
Medicine and health sciences	340	34%	
Physical sciences	21	2%	
Other	41	4%	
Region (derived from country)			
Africa	127	13%	
Asia	113	12%	
Australia and New Zealand	20	2%	
Europe	315	32%	
Latin America and the Caribbean	89	9%	
Middle East	19	2%	
North America	293	30%	

The majority of respondents (63%) indicated that they had been awarded the terminal degree in their field more than 10 years ago. More than three-quarters (78%) were from the fields of Biology and Life Sciences or Medicine and Health Sciences, with fewer than 100 responses from Ecology and Environmental Sciences, Engineering and Technology, Humanities and Social Sciences, and Physical Sciences. The majority of respondents (62%) were from North America or Europe, but a substantial number of responses were received from Africa (13%), Asia (12%), and Latin America and the Caribbean (9%).

Current methods sharing practices

Respondents were asked to indicate whether and, if so, how they had shared detailed methods information independently of their research findings in the past 5 years. Almost a quarter of respondents (n = 231; 23%) had not shared detailed methods information outside of their group or team (Fig. 1). The majority of respondents (n = 561; 56%) had privately shared detailed methods information with individual researchers upon request; respondents were not separately asked whether they had received such a request. A third (n = 327; 33%) had publicly shared detailed methods information in or on a separate publication or platform.

Figure 1 Researchers are sharing detailed methods information outside of their group or team, although private sharing is more common than public.

Percentages add to more than 100%, as respondents could select more than one response.

The prevalence of these methods sharing approaches varied across respondent career stage, field, and region. Respondents at later career stages appeared to be more likely to share outside of their group or team; public sharing, in particular, increased from 25% for respondents who had not yet been awarded their terminal degree to 29% at 0–5 years and 33% at 6–10 years (Table 2).

Table 2 Approaches to methods sharing by career stage.

Career stage	Only within
group or team (n)	% of responses	Privately shared (n)	% of
responses	Publicly
shared (n)	% of
responses	
Has not (yet) been awarded terminal degree	31	36%	43	49%	22	25%	
0–5 years since terminal degree	51	33%	77	50%	44	29%	
6–10 years since terminal degree	40	31%	60	47%	42	33%	
11–15 years since terminal degree	25	17%	92	61%	50	33%	
16–20 years since terminal degree	18	17%	60	57%	45	43%	
20+ years since terminal degree	66	18%	208	56%	123	33%	
Note:

Percentages add to more than 100%, as respondents could select multiple responses.

In exploring differences between fields with more than 100 respondents, Biology and Life Sciences was associated with more private sharing (62% vs. 54%) and less public sharing (31% vs. 34%) than Medicine and Health Sciences (Table 3). For fields with fewer than 100 respondents, Ecology and Environmental Sciences (37%) and Engineering and Technology (48%) were associated with higher than average public sharing. In exploring differences by region, public sharing was most common among respondents from Europe (37%) and North America (36%), and least common among respondents from Africa (23%) and the Middle East (16%) (Table 4).

Table 3 Approaches to methods sharing by field.

Field	Only within
group or team (n)	% of responses	Privately shared (n)	% of
responses	Publicly
shared (n)	% of
responses	
Biology and life sciences	92	21%	270	62%	137	31%	
Ecology and environmental
sciences	14	19%	44	59%	28	37%	
Engineering and technology	3	14%	10	48%	10	48%	
Humanities and social sciences	22	35%	30	48%	16	26%	
Medicine
and health sciences	86	25%	182	54%	114	34%	
Physical sciences	6	29%	13	62%	7	33%	
Note:

Percentages add to more than 100%, as respondents could select multiple responses.

Table 4 Approaches to methods sharing by region.

Region	Only within
group or team (n)	% of responses	Privately shared (n)	% of
responses	Publicly
shared (n)	% of
responses	
Africa	51	40%	57	45%	29	23%	
Asia	33	29%	59	52%	33	29%	
Australia and New Zealand	3	15%	12	60%	6	30%	
Europe	58	18%	183	58%	116	37%	
Latin America and the Caribbean	19	21%	48	54%	28	31%	
Middle East	10	53%	9	47%	3	16%	
North America	52	18%	185	63%	105	36%	
Note:

Percentages add to more than 100%, as respondents could select multiple responses.

Respondents who had publicly shared detailed methods information (n = 327), hereafter referred to as public sharers, were asked to identify the channels they had used to publicly share. Publishing a peer-reviewed protocol or methods publication (n = 168; 51%) and sharing in supporting information or as an appendix to a research article (n = 166; 51%) were the most common channels, followed by posting to a methods sharing platform (n = 112; 34%) and posting to a lab or project website (n = 79; 24%) (Fig. 2). A majority of public sharers who had published peer-reviewed protocols or methods publications (n = 107; 63%) reported that these publications were often or always published open access; however, this study focuses on public sharing rather than open sharing to avoid preemptively excluding paywalled publications or other outputs not under a formal open license.

Figure 2 Public sharers are most likely to share detailed methods information in channels connected to peer-reviewed publications.

Percentages add to more than 100%, as respondents could select more than one response.

In free-text responses, other channels for public sharing included generalist repositories like Figshare, Zenodo, and the Open Science Framework, as well as the code hosting service GitHub. Semipublic channels included teaching, conferences and workshops, and standard operating procedures used within specific organizations. Private channels spanned various forms of personal communication, including the hosting of lab visits.

Flaws were discovered in the design of a question about how frequently respondents engaged in different approaches to sharing (i.e., only within a research group or team; privately upon request; publicly in/on a separate publication or platform), and as a result responses to this question were excluded.

Attitudes toward methods sharing

To understand their attitudes toward existing norms in scientific publishing, respondents were asked to rate the extent of their agreement with a series of statements about the Methods sections of research articles in their field and, specifically, whether the information found therein is usually adequate to carry out certain research tasks. Almost three-quarters (n = 736; 74%) of respondents agreed or strongly agreed that the information found in Methods sections is adequate to evaluate findings (Fig. 3). A smaller proportion—less than half—agreed or strongly agreed that the information found in Methods sections is adequate to reproduce results (n = 410; 42%) or to reuse and extend the method in a different context (n = 461; 47%). Public sharers reported a lower level of agreement with all three statements (evaluate findings: n = 224/69%; reproduce results: n = 121/37%; reuse in different context: n = 132/41%) than non-public sharers (evaluate findings: n = 512/77%; reproduce results: n = 289/44%; reuse in different context: n = 329/50%). These differences were statistically significant (evaluate findings: p = 0.01; reproduce results: p = 0.03; reuse in different context: p = 0.01).

Figure 3 The information found in Methods sections is more widely regarded to be adequate for evaluating findings than for reproducing results or enabling reuse in a different context.

The extent of respondents’ agreement with the statements about the Methods sections of research articles in their field also varied by career stage, field, and region. For instance, respondents who had not yet been awarded their terminal degree appeared to show lower levels of agreement with all three statements (evaluate findings: 66%; reproduce results: 31%; reuse in different context: 41%) than respondents at other career stages (Table 5).

Table 5 Agree or strongly agree that the information found in Methods sections is adequate to carry out research tasks, by career stage.

Career stage	Evaluate
findings (n)	% of responses	Reproduce results (n)	% of
responses	Reuse in different context (n)	% of
responses	
Has not (yet) been awarded terminal degree	57	66%	27	31%	36	41%	
0–5 years since terminal degree	110	71%	72	47%	78	51%	
6–10 years since terminal degree	85	66%	49	39%	59	46%	
11–15 years since terminal degree	111	74%	63	43%	78	53%	
16–20 years since terminal degree	80	76%	38	36%	44	42%	
20+ years since terminal degree	292	79%	160	43%	166	45%	

In exploring differences between fields with more than 100 respondents, Biology and Life Sciences was associated with lower levels of agreement (evaluate findings: 70%; reproduce results: 32%; reuse in different context: 39%) than Medicine and Health Sciences (evaluate findings: 77%; reproduce results: 50%; reuse in different context: 53%) (Table 6). Among fields with fewer than 100 respondents, Engineering and Technology was associated with the lowest level of agreement for evaluating findings (71%) while Ecology and Environmental Sciences was associated with the lowest levels of agreement for reproducing results (42%) and reusing in a different context (42%). In exploring differences by region, levels of agreement were lowest among respondents from North America for evaluating findings (66%) and from Australia and New Zealand for reproducing results (26%) and reusing in a different context (32%) (Table 7). Levels of agreement were highest among respondents from Africa (evaluate findings: 81%; reproduce results: 66%; reuse in different context: 73%).

Table 6 Agree or strongly agree that the information found in Methods sections is adequate to carry out research tasks, by field.

Field	Evaluate
findings (n)	% of responses	Reproduce results (n)	% of
responses	Reuse in different context (n)	% of
responses	
Biology and life sciences	303	70%	139	32%	167	39%	
Ecology and environmental sciences	56	75%	31	42%	31	42%	
Engineering and technology	15	71%	12	57%	9	43%	
Humanities and social sciences	52	84%	30	48%	38	61%	
Medicine
and health sciences	264	77%	168	50%	180	53%	
Physical sciences	16	76%	12	57%	14	67%	

Table 7 Agree or strongly agree that the information found in Methods sections is adequate to carry out research tasks, by region.

Region	Evaluate
findings (n)	% of responses	Reproduce results (n)	% of
responses	Reuse in different context (n)	% of
responses	
Africa	103	81%	82	66%	91	73%	
Asia	86	76%	63	56%	65	58%	
Australia and New Zealand	15	75%	5	26%	6	32%	
Europe	240	76%	110	35%	135	43%	
Latin America and the Caribbean	71	80%	50	56%	50	56%	
Middle East	13	68%	7	37%	7	37%	
North America	193	66%	81	28%	96	33%	

Next, respondents were asked to rate the importance of publicly sharing detailed information about certain types of method: existing methods that are straightforward to apply and widely used, existing methods that are difficult to apply consistently and/or successfully, existing methods that have been modified or combined in a distinctive way, and novel methods that have not been reported before. A majority of respondents rated publicly sharing detailed information about all four types as important or extremely important (Fig. 4). Public sharers rated all four types as more important than non-public sharers, but this difference was statistically significant for only two of the types: existing methods that are difficult to apply (p = 0.02), and modified or combined methods (p = 0.02).

Figure 4 Publicly sharing detailed information about all types of method is perceived as important, with methods that are difficult to apply, have been modified, or have not been reported before rated more highly.

Respondents were also asked to rate the significance of certain barriers to publicly sharing detailed methods information. The modal response for all items was the response with the lowest scale value, “Not significant” (Fig. 5). The barriers rated as significant or highly significant by the largest proportion of respondents were “It takes too long to prepare detailed methods information in a way that would be useful for others” (n = 378; 40%) and “I would not know how or where to publicly share detailed methods information” (n = 332; 34%).

Figure 5 Barriers to public sharing of detailed methods information are perceived to be of low to moderate significance.

In open responses, the most frequently mentioned additional barrier was journal word limits for methods sections. Other barriers included treatment of methods as trade secrets in industry contexts and lack of recognition for methods publications, even when they are peer-reviewed. Some respondents also questioned whether documented methods information could adequately replace two-way personal communication, especially when the user of the information is unknown to its creator. For these respondents, the expectation of public sharing threatened to undermine what one described as “the interactive nature of science”.

Importance and satisfaction scores of methods sharing goals

Respondents were asked to rate the importance of eight goals related to publicly sharing detailed methods information, as well as their level of satisfaction with their ability to achieve these goals. As discussed above, these responses were then converted into numerical importance and satisfaction scores. The mean importance scores for these eight goals ranged from 64.5 to 84.3, and the mean satisfaction scores ranged from 58.4 to 73.0 (Table 8).

Table 8 Mean importance and satisfaction scores for goals related to public methods sharing.

	Importance	Satisfaction	
Goal	n	Mean score	SD	n	Mean score	SD	
Providing enough information for others to evaluate my findings	986	84.3	17.4	968	73.0	20.4	
Providing enough information for others to reproduce my results	982	82.9	19.3	960	70.1	22.0	
Providing enough information for others to reuse or extend the method I used	979	81.3	18.7	956	69.5	21.2	
Getting feedback to further improve the method’s efficacy	980	73.2	24.4	937	58.4	23.9	
Making the method easily and permanently discoverable	973	77.6	21.4	941	61.7	23.9	
Ensuring that the method is accessible to anyone	976	78.3	23.8	944	62.4	24.0	
Getting credit for my work in a way that can advance my career	976	64.5	30.7	914	60.6	23.7	
Helping my students to get credit for
their work in a way that can advance their careers	964	80.5	23.5	883	65.0	23.7	
Note:

The higher the score, the more important the goal or satisfied the respondent is with their ability to achieve it.

Goals related to the usability of the methods information for research tasks, including evaluating findings, reproducing results, and reusing or extending the method, scored highest for both importance and satisfaction. Goals related to making the methods information discoverable and accessible had moderate scores for both importance and satisfaction. Goals related to getting feedback and credit scored lowest for both importance and satisfaction, although helping students to get credit scored higher on both measures.

Public sharers rated all eight goals as more important than non-public sharers did. These differences were statistically significant (p < 0.03) for all but one of the goals, which related to making the methods information discoverable. Public sharers rated themselves as more satisfied than non-public sharers with their ability to achieve six of the eight goals. However, these differences were statistically significant for only one goal, related to the usability of the methods information to evaluate findings (p = 0.02).

Value of product or service features

Respondents were asked to rate the value of ten potential features of a product or service intended to enable the public sharing of detailed methods information. The features rated as essential by the largest proportion of respondents were “Availability of the methods information on an open access basis” (n = 481; 50%) and “Visibility of the methods information in scholarly indexes” (n = 418; 43%) (Fig. 6). All but one of the other features, “Personal invitation from the product or service provider,” were rated as valuable or essential by at least 50% of respondents.

Figure 6 Value of features of a product or service intended to enable methods sharing.

In open responses, other features that respondents reported they would like to see in a product or service intended to enable the public sharing of detailed methods information included linking to publications and other research outputs like datasets associated with the method, availability of a formal citation and digital object identifier, user-generated feedback or commenting, low or no cost, and ease of use. Respondents also expressed interest in what they described as “methodology search engines” or a “web of methods,” which would allow for comparison of similar methods and selection of the most appropriate one.

Discussion

The results of this survey show that, while Methods sections of research articles may be perceived as adequate for evaluating article-level findings, they are not widely perceived to be adequate for supporting a broader set of research tasks (see Fig. 3). This finding is in line with existing scholarship on barriers to reproducibility in scientific research (Open Science Collaboration, 2014; National Academies of Sciences, Engineering, and Medicine, 2019) and has practical implications for publishers and other stakeholders that seek to enable reproducibility and reuse. Steps can be taken to improve Methods sections, including removing word limits (Nature, 2013), clarifying expectations around textual novelty (Jia, Tan & Zhang, 2014), using reporting guidelines for specific study designs (Moher, 2018), and adopting guidelines for the use of shortcut citations (Standvoss et al., 2022). But the results of this study also support the move to share detailed methods information in a less article-centric way, as part of a broader evolution in scholarly communication (Van de Sompel et al., 2004; Lin, 2016).

The survey results show that private sharing is currently the most common approach to sharing detailed methods information, with a majority (56%) of respondents reporting having engaged in it. Elsewhere, private sharing has been found to be the dominant approach to sharing other outputs such as research data (Allagnat et al., 2019), and methods sharing platforms like protocols.io report that private protocols outnumber public ones (Protocols.io, 2023). However, the results also show that a substantial minority (33%) of respondents have publicly shared detailed methods information independently of their research findings. This study is one of the first efforts to quantify the rate of adoption of this Open Science practice from the researcher’s perspective. The fact that detailed methods information is being shared at all, even privately, can be seen as a positive feature of existing research culture. But, given the known problems around access and equity associated with providing outputs like research data privately upon request (Vanpaemel et al., 2015; Tedersoo et al., 2021; Gabelica, Bojčić & Puljak, 2022), relying on private sharing to make detailed methods information available is not optimal to support advances in research.

Among researchers who have publicly shared detailed methods information, the survey results document a diverse set of channels being used (see Fig. 2). The most widely used channels are those connected to peer-reviewed publications. Standalone methods publications are more discoverable than other approaches to methods sharing and can help authors to get credit for their work, but they are not always openly accessible and may be limited to more novel methods (Leist & Hengstler, 2018). Sharing detailed methods information in supporting information files or as an appendix, while convenient for authors and commonly used for other outputs like research data, has disadvantages in terms of discoverability and preservation (Pop & Salzberg, 2015). A substantial minority (34%) of respondents are, however, making use of dedicated platforms for sharing detailed methods information. A recent set of recommendations for improving methods reporting in the life sciences identified the use of such platforms as a best practice to be promoted by institutions, publishers, and funders (Batista Leite et al., 2023).

The results of the survey around barriers to methods sharing are more ambiguous. The most significant barrier was found to be “It takes too long to prepare detailed methods information in a way that would be useful for others.” This finding, of lack of time as a major barrier to open research practices, is in line with previous studies of data sharing (Digital Science, 2021) and code sharing (Hrynaszkiewicz, Harney & Cadwallader, 2021a) practices. However, responses to these survey items were clustered on the low end of the ordinal scale, with at least 40% of respondents rating each of the items either “Not significant” or “Slightly significant.” This proportion is considerably higher than the proportion of respondents that reported publicly sharing detailed methods information. It is not clear whether there are other important barriers to sharing that were not included in the survey questionnaire, or whether the identified barriers exert more of an influence in practice than respondents indicated in the context of survey completion.

The public sharer’s perspective

Respondents who had publicly shared detailed methods information were less likely to agree that the information found in Methods sections is adequate to carry out research tasks than respondents who had not publicly shared detailed methods information (see Fig. 3). One possible interpretation of this finding is that these respondents engage in public sharing because they have previously found the information in Methods sections to be inadequate for their needs. Further research is needed to test this hypothesis and establish its boundary conditions; if valid, it would point to the potential for channeling researcher frustration with missing or inaccessible methods information into a lasting commitment to public sharing. Public sharers also viewed various methods sharing goals as more important than non-public sharers. It is reasonable to assume that these attitudes precede and give rise to public sharing behavior, but it is possible that they are also shaped by the experience of public sharing, for instance due to positive feedback from users.

In other areas, differences between the perspectives of public sharers and non-public sharers were less clear. Statistically significant differences were found in the two groups’ views on the importance of publicly sharing detailed information about certain types of method—namely, existing methods that are difficult to apply and modified or combined methods. But for the limit cases of existing methods that are straightforward to apply and novel methods that have not been reported before, for which statistically significant differences were not found, there may be greater consensus about the importance of publicly sharing detailed information. Similarly, the lack of difference between public sharers and non-public sharers in terms of satisfaction with their ability to achieve methods sharing goals suggests that public sharers may be less uniquely satisfied with existing methods sharing tools, relative to other researchers, than they are convinced of the importance of public sharing.

Potential opportunities to support researchers

Today, many institutions, publishers, and funders encourage researchers to share all relevant outputs supporting their research findings. However, the generality of this expectation may leave researchers unclear about whether it applies to detailed methods information. Understanding the goals that researchers have related to methods sharing is therefore essential for evaluating the fitness for purpose of specific methods sharing solutions. The goals related to methods sharing that respondents rated as most important—those related to usability for research tasks including evaluating findings, reproducing results, and reusing the method in a different context—were also the goals that respondents reported they were most satisfied with their ability to achieve.

It is notable that the importance and satisfaction scores for these three goals were so closely clustered, given the larger gap in agreement levels about the adequacy of Methods sections to support these different research tasks (see Fig. 3). Perhaps while respondents perceive Methods sections as less than adequate to enable reproducibility and reuse, they are satisfied with their ability to support these use cases through other approaches to methods sharing—including private sharing. Similar results were reported in previous research on tasks related to data sharing and reuse, where researchers were on average satisfied with their ability to share their own research data but dissatisfied with their ability to access other researchers’ data (Hrynaszkiewicz, Harney & Cadwallader, 2021b). Further research may be needed to reconcile these views, since it seems that sharers’ level of satisfaction with the job they are doing is not matched by the satisfaction level of researchers seeking to make use of what has been shared. If such a mismatch does exist, then it could form the basis for education and advocacy efforts aimed at better aligning approaches to sharing research outputs with users’ actual needs.

Goals related to making methods information discoverable and accessible, as well as getting feedback and credit, were regarded as less important than the three goals related to usability. But respondents were also less satisfied with their ability to achieve these goals. This suggests that, while researchers may not be highly motivated to adopt new solutions targeting these goals, there may be opportunities to refine existing solutions with these goals in mind. For example, enhancements that make detailed methods information associated with published articles more visible could support researchers’ desire for discoverability. One such solution implemented at PLOS is a prominent “See the protocol” button next to the title of each published Lab Protocol, which links to the associated protocol on protocols.io. Elsewhere, Bio-Protocol has partnered with publishers including eLife on a Request a Protocol service, by which missing protocols can be requested by readers and, if sourced successfully, associated with the published article. Seventy-two percent of survey respondents reported that they saw an option for facilitated public sharing upon request as a valuable or essential feature of a methods sharing product or service.

Our exploration of variance in importance and satisfaction scores across segments also implies the need for tailoring messages to different audiences: for instance, while the mean importance score for all respondents for “Getting credit for my work in a way that can advance my career” was 64.5 (see Table 8), the mean importance score for respondents who had not (yet) been awarded a terminal degree or were awarded it fewer than 10 years ago was 77.7. This suggests that researchers at earlier stages of their careers may be more drawn to methods sharing solutions that enable getting credit for their work. Indeed, it is notable that the first Lab Protocol to be published at PLOS ONE (Cerasoni, 2021) was authored by a PhD candidate.

One objective of this study was to better understand and, ultimately, improve the fit between PLOS’s Lab Protocol article type and researcher needs. The high value that respondents placed on features associated with Lab Protocols (see Fig. 6), including open access, visibility in scholarly indexes, and peer review, suggests that the design of the article type is generally aligned with what researchers value in a methods sharing solution. However, given the proportion of respondents (34%) who cited “I would not know how or where to publicly share detailed methods information” as a significant barrier to methods sharing, it is likely that more needs to be done to build awareness about this article type and others like it. Methods sharing need not always take the form of a peer-reviewed publication, but increasing awareness of this format among researchers could help them better achieve some of their research goals.

Limitations

Survey respondents from Europe and North America were overrepresented relative to their share of the global researcher workforce (UNESCO, 2021b), as were respondents at later stages of their careers. Coverage of fields outside of the life and health sciences was limited, such that samples for fields including Ecology and Environmental Sciences, Engineering and Technology, Humanities and Social Sciences, and Physical Sciences were too small to be considered representative of these research communities.

The study’s recruitment strategy relied heavily on PLOS authors, who are encouraged to share detailed methods information as part of any submission, and protocols.io users, who share detailed methods information on the platform. This approach was taken to help ensure a sufficiently large sample of public sharers, as defined above, to draw meaningful conclusions. However, it likely also biased the respondent base toward higher rates of methods sharing.

The survey questionnaire did not offer a definition of “method” or of “detailed methods information,” although it did provide respondents with the examples of step-by-step instructions and troubleshooting for points of failure. This approach was used to include the diversity of methods—from qualitative to quantitative and theoretical to experimental—used by researchers across the disciplines, as well as the various forms and names that what we refer to as detailed methods information can take. However, a small number of respondents indicated in open responses that they found the absence of a definition to be confusing; our data should be interpreted with this empirical diversity in mind.

Conclusions

The findings of this study explore researchers’ attitudes and behaviors with respect to methods sharing, and how those vary across career stage, field, and region. The results also establish a useful baseline for the adoption of methods sharing as an Open Science practice, which can be tracked over time as researcher attitudes evolve and as new methods sharing solutions become available. This study’s survey-based methodology might be usefully complemented by large-scale automated screening of research outputs to measure observable rates of sharing, which would make it possible to achieve more uniform coverage across fields of research and to compare attitudinal data from surveys with quantified observation of practices. Previous work in this vein has examined particular elements of methodological rigor and transparency, such as resource identifiability (Menke et al., 2020). Recently, PLOS developed an approach to measure public sharing of protocols as part of its Open Science Indicators initiative (Public Library of Science, 2023; LaFlamme, 2023). Such efforts could be paired with deeper dives into particular communities and the political, economic, cultural, and technological issues they face in relation to methods sharing, as with emerging research on the sharing of computational workflows (Goble et al., 2020).

While this study took public sharing of detailed methods information as its primary focus, the results point to private sharing as a practice of interest in its own right. Little, if any research has been carried out to date on requests for detailed methods information (but see Midway et al., 2022). What role, for instance, do social and professional networks play in whether or not a request is fulfilled? What channels for communication are regarded as appropriate? And what strategies might be most effective for turning private sharers into public sharers, leveraging their existing willingness to share in a way that would distribute the benefits to science more widely and equitably? Compiling a corpus of requests and their outcomes, while posing practical challenges, could open up multiple lines of inquiry.

Finally, as suggested above, research on methods sharing is likely to be more impactful if it is accompanied by research on how and under what conditions detailed methods information gets put to use. Research on other Open Science practices has arguably been marked by a disparity in the attention paid to sharing and reuse, respectively (Wallis, Rolando & Borgman, 2013). Meta-research on methods sharing has the opportunity to close this gap and to experiment with novel research designs that illuminate how methods travel and transform—or are held constant—over time. A recent ethnographic study of reading practices in a lab-based journal club (Klein, 2023) showed how attention to embodiment and the material environment can reveal engagements with the methods of others that are situated and social rather than abstract and technical in nature. How such engagements can be supported and scaled in the service of high-quality research is a question that remains to be answered.

The authors would like to thank Emma Ganley, Gabriel Gasque, Veronique Kiermer, and Lenny Teytelman for comments on a draft of this manuscript. Beruria Novich contributed to the development of the survey questionnaire, and Ross Gray contributed to the statistical analysis.

Additional Information and Declarations

Competing Interests

Author Contributions

Human Ethics

Data Availability

Marcel LaFlamme, James Harney, and Iain Hrynaszkiewicz are employees of PLOS.

Marcel LaFlamme conceived and designed the experiments, performed the experiments, analyzed the data, prepared figures and/or tables, authored or reviewed drafts of the article, and approved the final draft.

James Harney conceived and designed the experiments, analyzed the data, authored or reviewed drafts of the article, and approved the final draft.

Iain Hrynaszkiewicz conceived and designed the experiments, authored or reviewed drafts of the article, and approved the final draft.

The following information was supplied relating to ethical approvals (i.e., approving body and any reference numbers):

The Heartland Institutional Review Board granted approval of this study (07092023-494).

The following information was supplied regarding data availability:

The survey questionnaire and deidentified data are available at Figshare: LaFlamme, Marcel; Harney, James; Hrynaszkiewicz, Iain (2022). Data from: A survey of researchers’ methods sharing practices and priorities. figshare. Dataset. https://doi.org/10.6084/m9.figshare.21334896.v2

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
