# Peer review of "A survey of researchers’ methods sharing practices and priorities"

_PeerJ, doi:10.7717/peerj.16731_

## Round 0.1 · original submission · Major Revisions

Thank you for your interesting and well-written manuscript. As you will see, two reviewers have looked at your work and made a number of comments for you to address. For each comment of theirs, and the additional comments I’ve added below, please provide a response explaining how you have changed your work in response to it or why you have not made changes for that particular comment.

1) I wonder if it would be helpful to use “questionnaire” when you refer to the form itself (rather than “survey instrument” on e.g. Line 75). For example, Line 81 talks about “The survey design” which could potentially refer to the questionnaire or the methods used to identify and invite potential respondents.

2) I appreciate that this point might also appear stylistic, but references to “Likert”, and particular “Likert scale” (e.g., Line 87), sometimes raise questions, including about the equal-spacing along some construct. Sometimes “ordinal item” or other options (see, for example, https://www.john-uebersax.com/stat/likert.htm for one perspective and some suggestions around naming) is useful to avoid such pedantry. I’ll leave this point entirely up to you.

3) I wonder if readers would be interested in knowing a little more about the pretesting stage (Line 95). In particular, how was this done (including who was involved) and how did this result in the questionnaire in [18] (and where did the questionnaire in [17] inform this—Lines 81–82)?

4) I was at times unclear whether you were describing the data in your sample (in which case proportions and means+SDs would be appropriate) or making estimates for the population (where CIs (or SEs) become appropriate, and p-values would be a useful addition in places). Given the survey design, I’m not presently convinced that you have a well-defined population to make inference about, but would be very happy to hear arguments to the contrary. This makes the CIs around proportions and means seem out of place to me. For the means, you could simply replace the CI half-widths with SDs; for proportions the SDs are defined in terms of the proportions. At the same time, comparing responses by groups can be enhanced by formal statistical testing, which you do at times, and I think some readers will look for non-overlapping CIs as evidence of differences (non-overlapping 95% CIs indicate statistical significance at approximately the 0.01 level and so the interpretation mentioned Reviewer #2 is more stringent than a conventional test at the 0.05 levels—note also that both Reviewers touched on the issue of multiple comparisons, which could be addressed by positioning this as an exploratory, hypothesis-generating study if that is your intention, or by more formal treatment of multiplicity, although you would need to think carefully about where you want to protect your family-wise error rate). While point-estimates, including proportions/means, can be questioned without a well-defined population and/or non-response (including non-completion here), differences are often be expected to be more robust. Two possibilities that occur to me would be to make the tables entirely descriptive (i.e., remove CIs from proportions in Tables 2–7 and replace CI for means with SDs in Table 8) or to do this and add a full set of formal statistical tests looking at comparisons of interest.

5) Related to the above, and as mentioned by both Reviewer #1 and Reviewer #2, a t-test unexpectedly appears in the Discussion (Lines 364–366) where I agree with them that the approach should have been covered in the methods and that the findings belongs in the results, but I’d like to add that actual p-values should be presented in all cases (perhaps with “p<0.001” for the smallest values). The magnitudes of these differences would seem more important than statistical significance and I think readers will appreciate seeing your interpretation about the practical significance of these results. Similar points would apply to Lines 371–376, 376–379, 385–389, 391–393, and perhaps elsewhere. Note that readers will want to know in the methods a little about the checks you used before applying these tests (do you have reasonable evidence that the populations are sufficiently normally distributed, and that variances are sufficiently equal across groups unless a Welch t-test was used? This should help to address a comment from Reviewer #1 about non-parametric alternative approaches.) Note also that a non-statistically significant finding (e.g., Line 393) warrants interpretation of the CI for the difference to see if this (in an inferential sense) rules out or fails to rule out practically important effects. This would be a reason in favour of retaining “statistically significant” and adding “of practical importance” or similar when appropriate (c.f. comment from Reviewer #1).

6) With Figures 3 and 4, I wonder if showing both public sharers and non-public sharers would help readers to appreciate the differences more than showing overall results and public sharer results. This is especially so given the discussions (e.g. Lines 362–363) comparing these groups.

·

Basic reporting

The manuscript investigates sharing practices of research methods
across different disciplines and researcher demographies and attitudes
based on a survey of nearly 1000 respondents. They assess how the
methods are shared, and what are the underlying motivations and
contexts where methods are being shared. The work highlights an
important and timely aspect of responsible research literature and
provides guidance for actors who strive to enhance transparency and
sharing of research methods.

The report is well-written and clear. Topical literature has been
cited appropriately. The manuscript is well-prepared.

Minor comments:

- Could "statistically significant" be written just "significant"; the
meaning is well-established in research.

- Table 1 might work well also in Figure format

- Figure 7 axes could indicate the units as well; remove "Plot of"
from the caption as redundant expression?

- Terminology: should we refer to "PhD candidate" or "PhD researcher"
instead of "PhD student"? Some universities in Europe have
implemented such change and it would better reflect the fact that
this career stage is a profession already.

- Consider citing EU open science policy and/or UNESCO recommendations
on open science.

Experimental design

The research question is well defined and relevant and fills an identified knowledge gap. The respondents are volunteers who decided to participate the survey after receiving online invitations. This may bring possible bias to the study population but the issues has been discussed appropriately.

1. Methods are described with sufficient detail and information to
replicate. However, I could not find the source code or reproducible
notebook for the data analyses of this manuscript.. this would
simplify exact replication and verification of the results and
improve transparency of the analyses.

2. Clear definition for "Methods" is not provided, and this is said to
be an intended design choice for the research. It can, however, have
a potentially remarkable impact on the data collection and
results. Introduction or Discussion could include more discussion on
qualitative / quantitative; theoretical / experimental; early
vs. late sharing (w.r.t the overall research cycle); workflows
vs. their individual components; and other possible divisions, and
how this might impact the interpretation of data.

3. The work could distinguish between public and open sharing, or at
least discuss this difference in more detail. It could be expected
that a substantial fraction of public sharers have also shared the
methods openly and recognizing this difference is relevant for this
type of work.

Validity of the findings

The underlying data, except the analysis source code or notebooks,
have been provided; the data are robust and collection procedure is
well reported. Conclusions are supported by the data.

- p. 15; Wilcoxon test might be better justified for this data than
t-test; is the significance robust to this change, and has multiple
testing been corrected?

Additional comments

5. It is interesting to note that only 33% of the respondents have
shared research methods publicly, even though many institution,
funder and published guidelines nowadays explicitly require or
recommend broad sharing of all relevant information (e.g. source
code and other methods). If not more researchers share methods, does
this imply that the current guidelines are not sufficient, or not
sufficiently monitored and enforced. Is more research needed on
this? What might be the discipline specific differences?

6. Much of the discussion has focused on open sharing of research data
and less on the sharing methods; is it possible to provide
suggestions or recommendations related to guidelines and policies on
(open or public) methods sharing?

7. Discussion section reports results and significances; ideally, the
utilized statistical tests would be reported in the Methods section,
observed differences including their significances and result
figures would be reported in the Results section, and Discussion
would then focus on qualitative interpretation and discussion and
avoid repeating statistical analysis details.

Reviewer 2 ·

Basic reporting

This article is well-written and relatively easy to read. It is clear in most places (see Line comments in general comments for areas of confusion). They authors describe in enough detail the issues associated with inadequately detailed methods and the background on the topic. The article is structured appropriately although the authors present a number of results, mainly statistical analysis, for the first time in the discussion. This should be moved to the results section, and statistical analysis described in the methods section.

Experimental design

The authors provide a relevant and meaningful goal to evaluate levels of current method sharing as well as barriers and motivation to sharing. This is a valuable contribution to the literature on the topic of open and reusable science. The survey used seems adequately designed, and the authors do a good job of outlining limitations. The only area that needs more description is the statistical analysis carried out. It's unclear how many statistical comparisons were made (only those presented? Or were other analyses done but not presented?).

Validity of the findings

Findings appear valid, and could be replicated with the provided survey instrument.

Additional comments

Line 136: Respondents had privately shared detailed methods with researchers upon request, but did the surveyors ask if respondents had received requests?
Line 201: Attitudes towards method sharing – interesting result that public sharers tended to agree less that methods sections were adequate. So do they share because they have found methods sections are inadequate? Or is their sharing independent of that? Just a thought
Line 271: Why convert the Likert scores in Table 8 to measures out of 100? Seems unnecessary, but I admit I’m unfamiliar with the presentation of Likert data. Presumably a Likert score of 5 would be a 100? And 4 would be 75? And so on? That could possibly be clarified. Also could you define satisfaction in more detail? I don’t understand what is meant to be satisfying – achieving the goal? Trying to achieve the goal?
Line 300: I’m curious if any of the open respondents described why they found certain products or features not valuable at all, such as open access.
Line 364: The results of this statistical analysis should be in the results section. In general, I’d like to see more statistical analysis in the results section. Some of the comparisons made seem like they would be non-significant based on the overlapping confidence intervals available in the tables. Unless specific formatting guidelines specify otherwise, please add the leading 0 before the decimal in p-values as well.
Line 375: Move stats to results. Same comment for the remainder of the discussion. Statistical tests used should be described in the methods. And given the large number of comparisons being made, it is worthwhile to consider using a multiple comparisons adjustment (although not the Bonferonni correction, as that is known to be too conservative).
Line 403: Could you provide a brief description of these quantitative thresholds and what they indicate? I’m not sure what it means that a goal falls below the marginal opportunities line. Worth describing in the methods. The paragraph from 400-403 should be moved to the methods and described in the results.
Line 441: Unless specific formatting guidelines specify otherwise, sentences typically should not start with a number – instead write out “Seventy two percent”.
Tables – Write out “Percentage” or just use %. I kept reading “%age” as “Percent age or responses” and was expecting an age.
Figures – add error bars to bar plots. Could be confidence intervals or standard error (or 2x standard error), whatever might be typical in the field, just describe in methods and/or figure legends.

---

## Round 0.2 · accepted · Accept

Thank you for your thoughtful revisions and responses our helpful reviewers' comments as well as my own. I am delighted to accept your very well-written manuscript and look forward to it generating discussion. Each time I re-read it, I'm left wanting to think more about the questions you raise and I hope that you will continue to work in this important area.

Reviewer 2 ·

Basic reporting

No comment - the authors have addressed the issues I raised to my satisfaction.

Experimental design

No comment - the authors have addressed the issues I raised to my satisfaction.

Validity of the findings

No comment - the authors have addressed the issues I raised to my satisfaction.